# Women's priorities towards ovarian cancer testing: a best–worst scaling study

Rebekah Hall ,[1] Antonieta Medina-Lara,[1] Willie Hamilton ,[2] Anne Spencer[1]

[1]Health Economics Group, University of Exeter Medical School, Exeter, UK
[2]Primary Care Diagnostics, University of Exeter Medical School, Exeter, UK

**Correspondence to**
Rebekah Hall;
rh591@exeter.ac.uk

## ABSTRACT

**Objective** To investigate the importance of key characteristics relating to diagnostic testing for ovarian cancer and to understand how previous test experience influences priorities.

**Design** Case 1 best–worst scaling embedded in an online survey.

**Setting** Primary care diagnostic testing in England and Wales.

**Participants** 150 women with ovaries over 40 years old living in England and Wales.

**Methods** We used best–worst scaling, a preference-based survey method, to elicit the relative importance of 25 characteristics relating to ovarian cancer testing following a systematic review. Responses were modelled using conditional logit regression. Subgroup analysis investigated variations based on testing history.

**Main outcome measures** Relative importance scores.

**Results** 'Chance of dying from ovarian cancer' (0.380, 95% CI 0.26 to 0.49) was the most important factor to respondents, closely followed by 'test sensitivity' (0.308, 95% CI 0.21 to 0.40). In contrast, 'time away from usual activities' (−0.244, 95% CI −0.33 to −0.15) and 'gender of healthcare provider' (−0.243, 95% CI −0.35 to −0.14) were least important to respondents overall. Women who had previously undergone testing placed higher importance on certain characteristics including 'openness of healthcare providers' and 'chance of diagnosing another condition' at the expense of reduced emphasis on characteristics such as 'pain and discomfort' and 'time away from usual activities'.

**Conclusions** The results clearly demonstrated items at the extreme, which were most and least important to women considering ovarian cancer testing. Differences in priorities by testing history demonstrate an experience effect, whereby preferences adapt over time based on evidence and experience. Acknowledging these differences helps to identify underlying barriers and facilitators for women with no test experience as well as shortcomings of current service based on women with experience.

## STRENGTHS AND LIMITATIONS OF THIS STUDY

⇒ This study adds to a very limited evidence base of studies assessing priorities towards diagnostic testing for cancer and specifically ovarian cancer.
⇒ Selection of included characteristics is based on a rigorous, published systematic review and used patient and public involvement to ensure the relevance to the target population.
⇒ Case 1 best–worst scaling is ideal for establishing the relative importance of a large number of characteristics and has been proven to be easier to complete and more effective than alternative methods such as ranking or ratings tasks.
⇒ A key limitation of the study relates to the representativeness of the sample. Due to the recruitment method, the sample is not fully representative of the population in key demographics including ethnicity and age distribution.
⇒ The lack of discrimination between lower scoring attributes may be reflective of genuine priorities. However, it is also possible that choice task construction and sample size were contributing factors.

high mortality rates associated with ovarian cancer and is an ongoing challenge globally.[3] For instance, in the UK, almost 60% of cases are diagnosed at stage III or IV where average 5-year survival is just 26.9% and 13.4%, respectively.[4] Improving diagnostic outcomes is multifaceted problem, however, delays in help-seeking on symptom onset and access to timely testing have been identified as challenges to earlier diagnosis.[5]

National guidelines for the investigation of suspected ovarian cancer in symptomatic women vary substantially between countries.[6] Existing tests include the CA125 blood test and imaging tests, most commonly a transabdominal/transvaginal ultrasound (TVUS) but also CT and MRI.[7] Variations in guidance represents uncertainty around the accuracy of existing test strategies, especially for the early investigation of symptoms. Evidence evaluating the performance of diagnostic tests in a primary care setting is very limited.[8] Furthermore, for recommendations to be effective, clinical guidelines must also consider the preferences of those offered

## INTRODUCTION

Ovarian cancer is the seventh most common cancer in women worldwide, with over 200 000 new cases and approximately 180 000 deaths annually.[1] Five-year survival rates for the disease are highly dependent on a number of factors including the patient's age, country of residence and tumour type.[2] Late-stage diagnosis contributes heavily to the

testing, particularly in healthcare systems emphasising shared-decision making.[9] Available tests differ substantially, not only in terms of performance but also service delivery and patient experience. Aspects of tests may be prioritised differently; for instance, patients may accept a lower levels of accuracy a less invasive test. Thus, it may be necessary to weight characteristics differently when considering the overall balance of benefits and harms.

To date, very little attention has been paid to the priorities of women facing ovarian cancer testing, particularly in a diagnostic setting. Existing studies usually focus on screening trials,[10] single test modalities[11] or single aspects of acceptability, such as pain.[12] To address the current evidence gap, we aimed to elicit preferences for 25 key characteristics ('items') of ovarian cancer testing, using best–worst scaling (BWS). This is a stated preference technique that has been demonstrated to provide higher predictive values than ranking or rating methods while being less cognitively demanding.[13] Understanding the importance of key characteristics allows aspects of greatest importance to be given increased salience in future decisions about testing, particularly in guideline revision.

Previous studies have demonstrated views may differ based on previous experience of the health event. We; therefore, examined the relationship between priorities around ovarian cancer testing and test experience.[14 15]

## METHODS

### Patient and public involvement in the research
The survey was shared and discussed with the Cancer Research UK-funded CanTest PPI lead, Margaret Johnson. Amendments in survey wording were made as a result of discussions.

### Best–worst scaling
We used case 1 BWS method to identify women's preferences for characteristics associated with diagnostic testing for ovarian cancer. This type of BWS aims to assess the relative importance of, or preference for, items based on the underlying principles of random utility theory.[16] BWS was initially developed in marketing but has been increasingly used in healthcare research for explorations of patient and stakeholder preferences.[16–18] In particular, case 1 BWS has been demonstrated to be an effective alternative to traditionally used ranking or ratings tasks when considering patient priorities.[19]

During BWS tasks, participants respond to a series of choice tasks presenting a subset of items and asked to select the 'best' or 'most important' and 'worst' or 'least important' item. Simultaneously examining items selected as 'most' and 'least' important provides greater information than examining the most important item alone. Analysing responses to choice tasks allows the underlying relative importance of items to be inferred and a ranking of included items to be established.[19]

### Identification of relevant items
Results demonstrate the importance of an item, relative to the other included item. It is therefore important to use a rigorous selection process to ensure the most salient characteristics are included in the experiment. We performed a systematic review of the preference-based literature to identify characteristics relating to cancer testing.[20] Potential characteristics were then narrowed down by the authors based on relevance to ovarian cancer testing in symptomatic patients using an iterative Delphi method process where exclusion required full agreement of the research team. In total, 25 characteristics were selected for inclusion (table 1).

### BWS task construction
A balanced incomplete block design (BIBD) generated using SAS V.9.4 was used to construct the BWS choice sets for the 25 items. BIBD designs ensure each item appears equally often, and pairwise comparisons between each of the items occurs equally across the design.[21] A final design (d-efficiency of 83.3%) consisting of 30 choice tasks each with five items was selected based on a trade-off between survey complexity and design efficiency. Each item appeared six times across the choice tasks and coappeared with remaining attributes once throughout the experiment. The position of the item within choice tasks was optimised such that items were listed in 1st–5th position equally and the order of tasks was randomised across participants to control for any order effects. Each participant completed all 30 tasks. An example of a BWS task is shown in figure 1.

### Questionnaire
The questionnaire was developed and collected using LimeSurvey, an online survey platform. The survey consisted of four stages as follows: (1) Sociodemographic questions, for example, age, education, employment status; (2) BWS questions (including a warm-up task); (3) Task feedback questions, for example, task difficulty and (4) Background questions relating to health-related characteristics, for example, self-reported health, testing history, current and desired in medical decisions.

Question framing and answer categories for stages 1 and 3–4 were copied or adapted from established national surveys. Early piloting with five women suggested the survey would take 30 min to complete. Given the length of the survey and online administration method, three attention checks following the instructional manipulation format were embedded (eg, 'Select 'very important' to indicate you are paying attention').[22] Respondents who failed all three attention checks were removed from the analysis. A full version of the questionnaire is available on request.

### Participants
A sample size of 150 women was estimated to be required. Participants were recruited via Prolific, an online platform for researchers conducting social science experiments. Results from this platform have been shown to be of comparable or better quality than university research lab registers and have been used widely within hundreds

**Table 1** Characteristics included in the BWS study (wording and descriptions are identical to those shown to respondents during the survey)

| | Characteristic and definition |
|---|---|
| 1 | Test sensitivity<br>Chance that the test will miss cancer in a patient who actually does have the disease (false-negative result) |
| 2 | Chance of dying from ovarian cancer<br>How much having the test decreases the chance of dying from ovarian cancer |
| 3 | Choice of appointment time<br>Whether a person can choose an appointment time or if the appointment time is assigned by the healthcare provider |
| 4 | Who explains the results<br>Type of healthcare provider who explains the test results, for example, nurse, doctor, etc. |
| 5 | Pain and discomfort<br>The level of pain and/or discomfort experienced during and after the test |
| 6 | Notification of negative test results<br>Whether you are contacted if your results are normal |
| 7 | Chance of diagnosing another condition<br>If symptoms are not caused by cancer, the chance the test can identify other reason for the symptoms |
| 8 | Pretest support<br>Level of support received before having the test describing what will happen during the test and what the results might show |
| 9 | Test procedure<br>What having the test will involve. For ovarian cancer this could be a blood test or an transvaginal ultrasound (internal ultrasound of the reproductive organs) |
| 10 | Staff attitude<br>How the healthcare provider treats you while conducting the test |
| 11 | Post-test support<br>Level of support received after getting the results of the test relating to the meaning of your results and what will happen next |
| 12 | Time away from usual activities<br>The total amount of time spent having the test instead of doing your usual daily activities |
| 13 | Test specificity<br>Chance of unnecessary further invasive testing (false-positive result) |
| 14 | Travel time<br>The total amount of time spent travelling to and from the test |
| 15 | Time to notification of results<br>The length of time it takes to hear the results after having the test |
| 16 | Openness of healthcare providers<br>How open healthcare providers are with their thoughts about the cause of your symptoms and the tests they recommend |
| 17 | No of follow-up tests<br>How many additional tests are needed to confirm a diagnosis |
| 18 | Chance of an inconclusive result<br>The chance the results are unclear and the test would need to be repeated after a waiting period |
| 19 | Out of pocket costs<br>How much it will personally cost a person to have the test, for example, travel costs, childcare costs, time off work, etc. The cost of the test is covered by the NHS |
| 20 | Gender of healthcare provider<br>Gender of the staff member giving you the test |
| 21 | How test results are returned,<br>for example, in person, phone, letter |
| 22 | Test location<br>Where the test takes place |
| 23 | Test duration<br>The length of time spent having the test |
| 24 | Information included with the invitation<br>The level and type of information received about the test |
| 25 | Waiting time for the test<br>How long a person has to wait to have the test after being referred by their GP |

BWS, best–worst scaling; GP, general practitioner; NHS, National Health Service.

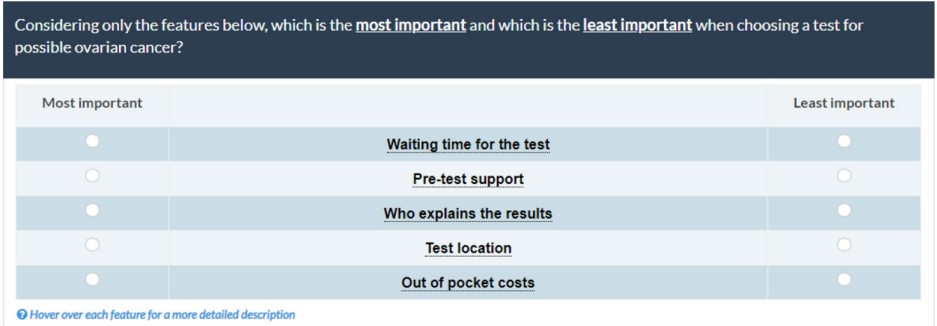

**Figure 1** Example of a choice task.

of published studies across disciplines.[23 24] Our study focused on the priorities of women over 40 (no upper limit), the group most likely to be offered testing.[25] Participation was also limited to those living in England and Wales: no other limitations were applied. Respondents completed an electronic consent form before completing any questions.

### Statistical analysis

Descriptive statistics relating to sociodemographic and health-related characteristics of the sample were summarised.

Best–worst responses were initially analysed using the counting approach; whereby the number of times an item was picked as 'least important' is subtracted from the number of times it was chosen as 'most important'. Each item appeared six times across all tasks, meaning best–worst scores could range from −6 to +6 at the individual level. Individual scores were aggregated and standardised to calculate an overall mean score for the sample ranging from −1 to +1 (Scores were standardised using the following equation: population-level best–worst score/(number of times item appeared × total sample size)). A score close to +1 indicates an item is highly influential, whereas item with scores near to −1 demonstrate much less relevance.

Next, conditional logit regression using dummy-coded items was used to model responses. 'Time away from usual activities' was identified as the least important attribute during counting analysis and was omitted from the model and used as the reference item. As a result, all parameters were expected to be positive. To aid interpretation, we rescaled conditional logit coefficients using standardised ratio scores where all scores sum to 100.[26 27] CIs for relative importance scores were estimated using the delta method. All analyses were performed using Stata V.17.[28]

In both the counting and conditional logit analysis, scores relate to the relative importance of attributes (ie, relative importance scores). In other words, differences between scores are meaningful whereas absolute values are not.

### Subgroup analysis

Subgroup analysis was performed to explore the relationship between previous testing and preferences for future

testing. Experience was captured by two questions that asked whether women had previously: (1) undergone testing for ovarian cancer or (2) received a transvaginal ultrasound for any reason. Subgroup analysis was based on conditional logit results and differences in relative importance scores between groups were tested using unpaired t-tests. Finally, heteroscedastic logit models were estimated to investigate whether differences between subgroup were attributable to scale differences (ie, differences in error variance between subgroups) or due to a genuine difference in preferences.

### RESULTS

In total, 159 women responded to the survey. The average response time for the questionnaire was 29 min 51 s. Four submissions were incomplete, two were removed due to failing all attention checks, and a further three responses were removed due to incorrect completion of the best–worst section of the survey, resulting in 150 responses for the final analysis. Respondents varied substantially in how difficult they found the best–worst portion of the questionnaire with 42% (63/150) reporting it easy/very easy but 38% (57/150) finding it difficult/very difficult. There were no significant differences in BWS responses between those who found the task difficult versus those who did not.

### Demographics

The demographics of the sample are presented in table 2. The age ranged from 40 to 87 years old with a mean of 51.4 (SD=9.1). Most participants were white (120/150; 80%), married/in a relationship (97/150; 65%) and employed (78/150; 52%).

Most women perceived their risk of cancer as low-average (128/150; 85%) and ovarian cancer-related anxiety was generally low-moderate among respondents (116/150; 77%). Overall, 50 women (33%) reported previously undergoing a TVUS for any reason. Forty (27%) women reported being previously tested for ovarian cancer, with CA125 blood test being the most common test.

Crucially, when asked, 127/150 women (89%) stated they wished for a great deal/a lot of involvement in decisions relation to their own care but only 34/150 (23%)

**Table 2** Descriptive characteristics: sociodemographic

| Characteristic | n (%) |
|---|---|
| Age | |
| Mean (SD) | 51.4 (9) |
| Range | 40–87 |
| Ethnicity | |
| White | 120 (80) |
| Asian | 8 (5) |
| Black | 3 (2) |
| Mixed | 3 (2) |
| Other | 9 (6) |
| Not reported | 7 (4) |
| No of children, mean (SD) | 1.3 (1) |
| Relationship status | |
| Married | 75 (50) |
| In a relationship | 22 (15) |
| Single | 19 (13) |
| Divorced/separated | 26 (17) |
| Widowed | 6 (4) |
| Not reported | 2 (1) |
| Level of education | |
| GCSE | 37 (25) |
| A-level/college | 25 (17) |
| Undergraduate | 41 (27) |
| Postgraduate | 35 (23) |
| No qualifications | 1 (1) |
| Other | 9 (6) |
| Not reported | 2 (1) |
| Employment status | |
| Full-time | 47 (31) |
| Part-time | 32 (21) |
| Self-employed | 23 (15) |
| Not employed | 11 (7) |
| Retired | 14 (9) |
| Other | 18 (12) |
| Not reported | 5 (3) |

GCSE, General Certificate of Secondary Education.

currently felt this was achieved, with 17/150 (11%) respondents felt unable to be involved in medical decisions at all. Further details are found in online supplemental appendix 1.

## Best–worst results
### Counting analysis
BWS results are presented in online supplemental appendix 2. Scores were bound between –1 and 1. Scores tending towards the extremes of the scale would imply homogeneity across respondents and consistency between responses across questions on an individual level. Importance scores ranged from –0.224 to 0.380, suggesting high

levels of heterogeneity in preferences regarding test characterisics across respondents (see online supplemental appendix 3).

Overall, 'chance of dying from ovarian cancer' (0.380, 5% CI 0.26 to 0.49) was most important to women when considering ovarian cancer testing, followed by 'test sensitivity' (0.308, 95% CI 0.21 to 0.40). Conversely, 'time away from usual activities' (–0.244, 95% CI –0.33 to –0.15) and 'gender of healthcare provider' (–0.243, 95% CI –0.35 to –0.14) were considered least important to women when facing diagnostic testing and were statistically indistinguishable from each other.

### Conditional logit analysis
The results for the conditional logit model are shown in online supplemental appendix 2. The order of importance remained consistent across the two analysis methods and the estimates were highly correlated (online supplemental appendix 4).

All items had a positive coefficient and most were statistically significant at the 95% level, confirming the relative

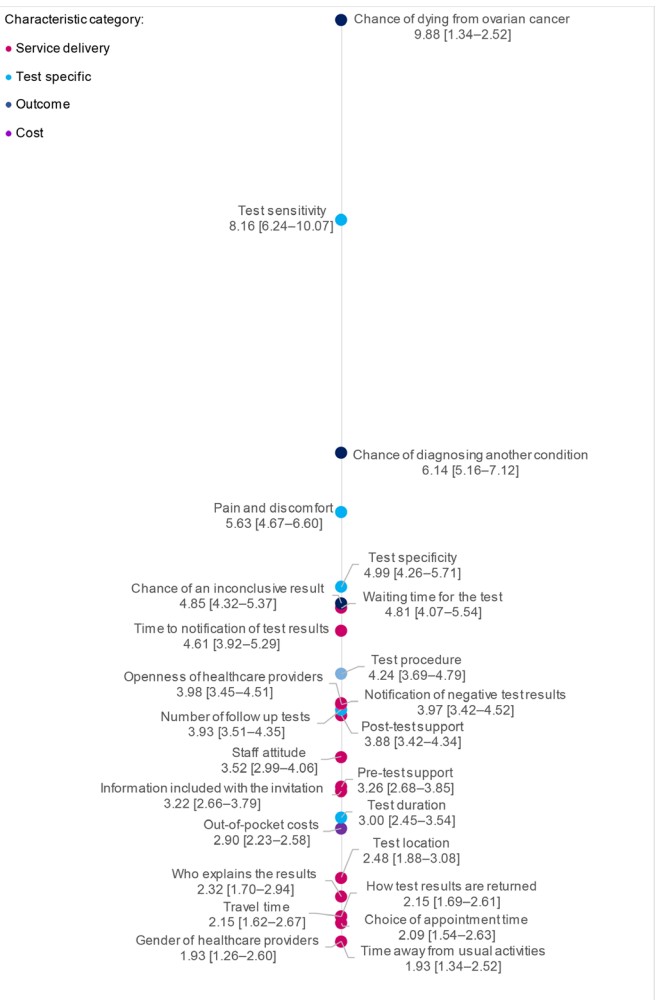

**Figure 2** Best–worst scaling results. The distance between attributes is a spatial representation of the difference in relative importance between attributes on the latent importance scale.

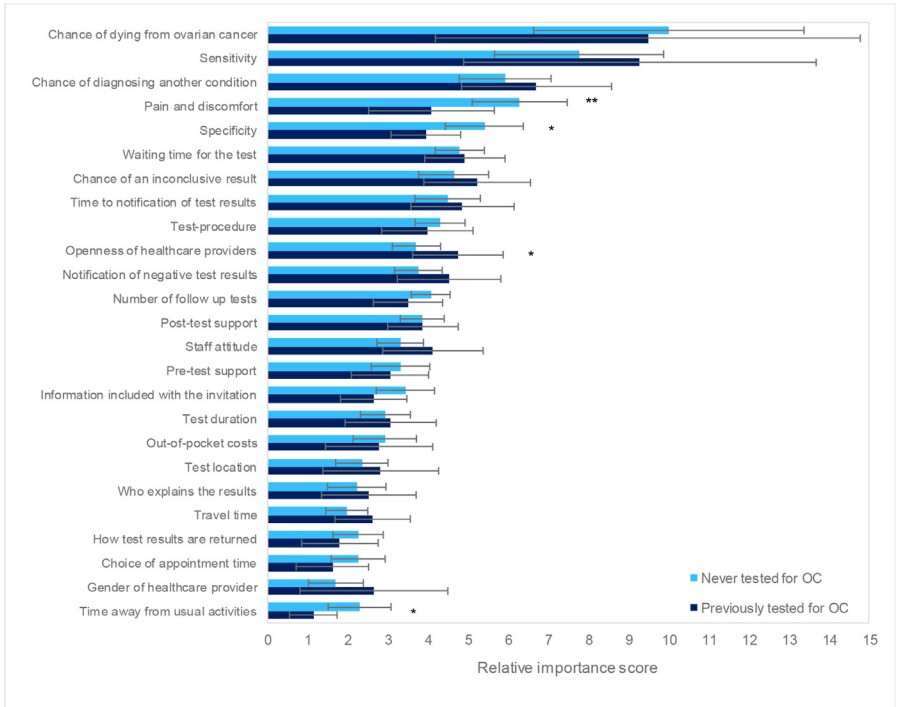

**Figure 3** Subgroup analysis results comparing importance scores between those previously tested for ovarian cancer (OC) (n=40) and those who have never been tested (n=110). Error bars represent 95% CIs. Significance of differences between subgroups: *p<0.10, **p<0.05, ***p<0.01

importance of all attributes compared with 'time away from usual activities'. Non-significant items were those with the lowest importance (eg, 'gender of healthcare providers' and 'how test results are returned'), suggesting a clustering effect towards the bottom of the importance scale.

Figure 2 shows the relative importance scores associated with the conditional logit estimates. The distance between each item is a spatial visualisation of differences in relative importance. Importance scores ranged from a maximum of 9.88 (95% CI 7.04 to 12.72) for 'chance of dying of ovarian cancer' to a minimum of 1.93 (95% CI 1.34 to 2.52) for 'time away from usual activities' and Gender of healthcare providers. Indicating 'chance of dying from ovarian cancer' was approximately five times more important to respondents. The most important characteristics to respondents were clear and distinct: however, spatial visualisation demonstrated grouping of attributes towards the centre and bottom of the scale. Groupings were distinct to other items but differences in importance between items within clusters is less distinguishable. In general, results demonstrate a clear prioritisation of outcome (dark blue dots) and test specific characteristics (light blue dots) whereas service delivery characteristics (pink dots) were consistently less important to respondents.

### Subgroup analysis
Women previously tested for ovarian cancer placed significantly lower importance on 'pain and discomfort' in comparison to test-naïve individuals. There was also

evidence to suggest 'time away from usual activities' and 'test specificity' were less prioritised by previously tested women while 'openness of healthcare providers' and 'test specificity' appeared marginally more important to those with test experience (figure 3).

For women who had previously undergone a TVUS, 'chance of diagnosing another condition' and 'chance of dying from ovarian cancer' were significantly more important compared with those with no test experience (figure 4). Alternatively, women who had never been tested appeared to place higher value on 'information included with the invitation' and 'choice of appointment time', although the overall importance of such attributes remained relatively low.

In both instances, results from heteroscedastic logit models demonstrated no differences in scale between subgroups indicating observed differences were attributable to differences in priorities as opposed to differences in response variance between groups with different prior experience (online supplemental appendix 5).

### DISCUSSION
#### Summary of main findings
This is the first study to investigate the priorities of women relating to diagnostic testing for ovarian cancer. The results of this study highlight the importance yet current neglect of incorporating the preferences of patients into medical testing decisions. When asked, almost 90% (134/150) of respondents wanted heavy involvement in

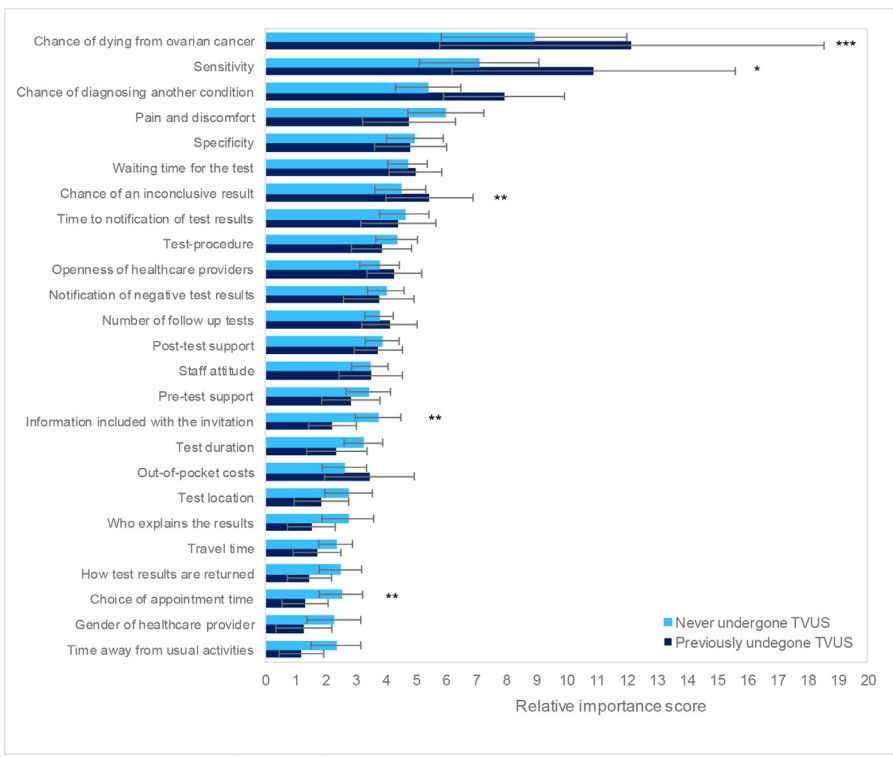

**Figure 4** Subgroup analysis results comparing importance scores between those who have previously undergone a transvaginal ultrasound (TVUS) for any medical reason (n=50) and those who have not (n=100). Error bars represent 95% CIs. Significance of differences between subgroups: *p<0.10, **p<0.05, ***p<0.01

medical decisions: however, less than a quarter of women currently felt able to do so.

The results are particularly useful in identifying items at either extreme of the scale, clearly demonstrating the characteristics that are most and least important to women considering ovarian cancer. Low concurrence between items and high levels of heterogeneity meant discrimination between mid-range items was less clear. Overall, 'chance of dying from ovarian cancer', 'test sensitivity' and 'chance of diagnosing another condition' were the most important characteristics when considering testing for ovarian cancer. Oppositely, 'time away from usual activities' and 'gender of healthcare provider' were the least important factors. The priorities of previously tested women were generally similar to women who had never undergone testing, but there were a few key differences between these two groups suggesting experience is an important determinant of priorities.

### Results in the context of published literature
The prioritisation of outcome and test-specific characteristics and the relative lack of importance placed on service delivery characteristics is reflective of findings from the wider literature examining preferences towards cancer screening and diagnosis.[20 29–32] The relative importance of 'openness of healthcare providers' is echoed by qualitative research exploring the diagnostic experiences of women with ovarian cancer where shortcomings in doctor–patient communication was a recurring theme. In particular, patients raised concerns about doctors'

willingness and ability to openly communicate and share information.[33 34] Decreased importance of particular characteristics to tested women may suggest these aspects of testing are already being achieved/acceptable within the current system. For example, the decreased importance of 'pain and discomfort' for both women who have been tested for ovarian cancer and women who have previously received an ultrasound (NS) is expected given the finding that almost 80% of women who had previously undergone a transvaginal ultrasound experienced little to no pain.[12]

Existing studies examining differences in preferences towards cancer testing between previously tested and untested individuals have typically focused on colorectal cancer screening, but also found evidence of statistically different preferences between the two groups.[35 36] Hol et al[35] found previously screened individuals displayed stronger and more positive preferences towards different screening modalities and shorter screening intervals than unscreened counterparts. In a similar study, van Dam et al[36] found limited differences between people with and without screening experience (mortality risk reduction only), however, preferences across previously screened individuals differed significantly based on the particular screening modality received. These results are evidence of an experience effect, via 'status quo bias', where individuals place a higher value on goods or services that are more familiar.[37]

## Strengths and limitations

BWS provides a straightforward method for capturing the priorities of women. The method is preferable to ranking/rating tasks due to its ability to measure the importance of large numbers of items while limiting complexity and cognitive burden.[38]

The survey length and high number of items represent a potential limitation. In total, 38% of respondents reported the task as difficult/very difficult. Despite this, drop-out rates were low (3% (4/150) of people began the survey but did not complete) and there were no differences between responses according to reported task difficulty.

Poor discrimination between lower scoring items may be reflective of genuine equality in priorities, however, it is also feasible that the choice task construction was a contributing factor. For example, the ability to detect differences will be affected by the number of pairwise comparisons between attributes. Selection of a BIBD with more than one pairwise comparison between each item may have increased the explanatory power of the study, of course this requires a trade-off with survey length and sample size requirements where blocking is required. Similarly, a larger sample size may have increased the explanatory power and also allowed for more complex statistical investigation of preference heterogeneity. To our knowledge, there is currently no accepted guidance on sample size requirements within object case BWS, although theories from the closely related methods, such as discrete choice experiments may be transferable.[39]

Finally, due to the recruitment method, the sample is not fully representative of the population in key demographics including ethnicity and age distribution. Further research is needed to understand whether results are generalisable to a wider population.

## Implications for practice and future research

The findings of this study offer useful insights into potential barriers and facilitators of undergoing testing in a timely manner. Interestingly, characteristics involving an element of risk or uncertainty dominated the top-ranking positions. However, how to best explain complex aspects of test performance such as sensitivity and specificity to patients is a clear challenge.[40]

Differences in importance between women with and without test experience suggest that priorities are continuously adapted based on evidence and experience gained over time. When considering policy decisions, it is therefore important to carefully consider whose views should be prioritised—a long-standing debate within the field of health technology assessment.[41] Arguably, it is important to consider both perspectives; priorities of women with test experience help to identify unmet or inadequate aspects of current service provision, whereas preconceived views of testing-naïve women may reveal underlying barriers and facilitators of testing since the initial decision to undergo testing is based on these pre-existing judgements. In both instances, mismatches in priorities

and practice may lead to delays in seeking help or testing for future symptoms.

High levels of heterogeneity between individuals highlight the importance of a personalised approach to patient interactions throughout the diagnostic process. Patients are likely to have different concerns and priorities during this time, however, the importance of patient-input and shared decision-making appears to be less prioritised in diagnostic settings compared with decisions regarding cancer screening and subsequent treatment where preferences have been studied more extensively.

## CONCLUSION

Preferences towards diagnostic testing have been under-explored to date. Understanding what matters most to patients may reduce anxiety around testing, facilitate earlier help-seeking behaviour and improve patient satisfaction. This study highlights that test sensitivity and mortality impacts are the most important factors to patients facing ovarian cancer testing. However, results varied significantly across individuals demonstrating the need for an individualised approach to consultations regarding diagnostic care. Our results can help inform policies and diagnostic guidelines designed to encourage earlier help-seeking behaviour as well as help to evaluate the patient-friendliness of emerging test strategies for suspected ovarian cancer.

**Acknowledgements** The authors thank Margaret Johnson, CanTest PPI Panel lead, for providing feedback during the development of this study.

**Contributors** All authors contributed to the study conception and design. RH and AM-L conducted data analysis. RH prepared the first draft of the manuscript and AS, AM-L and WH edited and approved the final version. RH is the guarantor for the study.

**Funding** The authors have not declared a specific grant for this research from any funding agency in the public, commercial or not-for-profit sectors.

**Competing interests** None declared.

**Patient and public involvement** Patients and/or the public were involved in the design, or conduct, or reporting, or dissemination plans of this research. Refer to the Methods section for further details.

**Patient consent for publication** Not applicable.

**Ethics approval** This study involves human participants and was approved by University of Exeter UEBS-REC eUEBS003725. Participants gave informed consent to participate in the study before taking part.

**Provenance and peer review** Not commissioned; externally peer reviewed.

**Data availability statement** Data are available on reasonable request. Data are available on reasonable request. Please contact the corresponding author with any enquiries.

purpose, provided the original work is properly cited, a link to the licence is given, and indication of whether changes were made. See: https://creativecommons.org/licenses/by/4.0/.

**ORCID iDs**
Rebekah Hall http://orcid.org/0000-0003-1804-993X
Willie Hamilton http://orcid.org/0000-0003-1611-1373

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
