## [Reviewer comments · BMJ Open]

ARTICLE DETAILS

TITLE (PROVISIONAL)	Women's priorities towards ovarian cancer testing: A best-worst scaling study
AUTHORS	Hall, Rebekah; Medina-Lara, Antonieta; Hamilton, Willie; Spencer, Anne

VERSION 1 – REVIEW

REVIEWER	Bridges, John F. P. Ohio State University
REVIEW RETURNED	03-Mar-2022

GENERAL COMMENTS	This study represents a solid application of best-worst scaling to assess patients values towards a diagnostic test. These approaches have been increasingly valued by decision makers and have been highlighted by large public-private partnerships as constituting best-practices (MDIC, IMI-PREFER etc). I have some comments as I read the manuscript. Overall, I think this is the start of a great paper and is well written, but I think the authors could do a better effort in describing and justifying their methods – and maybe do a better job on the methods and reporting. This should be an easy fix. The strengths and weakness of the study are somewhat trivial and highlight concepts that done really impact this study. For example, the authors mix concepts of preferences (like predicting market uptake) with the values/priorities elicitation process inherent in a BWS. Given this is a values/priorities elicitation, issues like hypothetical bias don't apply. It would be great for the authors to focus on the substantive and practical issues associated with this application. In the first paragraph of the introduction, which is a little terse, it would be good to know a little more about the personal and society burden of ovarian cancer. Section 2.1 is a nice background, but could do with more referencing to let the reader know that this is often used method and supported by decision makers. Section 2.2 is title experimental design, but is more about identification of the factors, then experimental design – maybe this is best as two sections. Also, it is normal to call these objects, not attributes, so as not get confused with preference studies. It would be nice for the experimental design section to have some references and to more rigorously test the characteristics of the design. “Patient and participant engagement” – this is great, but maybe need to be earlier in the methods section. It might be nice to add some references to support the standards and impetus for doing this. Also,
---

	what impact did these engagement efforts have? Analysis – MaxDiff is one of the ways to analyze this data, but there are others. Why was this approach chosen? This is then followed by a discussion of a simple Best minus worst scale – which is also good, but different. I think a more rigorous discussion of analysis (both options and decisions) is warranted. The results don't seem to present MaxDiff scores? Figure 2 is wonderful – but needs more explanation. What is on the Axis (I know, but the average reader may not)? What does it mean to be a cluster here? What are the hypotheses being tested? I think a more rigor methods section is need to explain this. Figure 3 is important, but hard to read. Also, if this was a MaxDiff or sequential best worst, then a ratio scale off importance could be made. This might really help here. Does this need to be on a bidirectional scale or unidirectional scale? Could the differences between the groups be scale (internal consistency within the groups). Appendix 2 is nice, especially as several decision makers want heterogeneity analysis (e.g. FDA/CDRH). This said, it is not formatted great. General discussion is good, but I would like to see the methodological issues addressed first. They are many easy fixes here and this could be a great paper with some more robust methods and clearer presentation of the results.
--	---

REVIEWER	Al-Aqeel, Sinaa recommended King Saud University , Clinical Pharmacy
REVIEW RETURNED	06-Mar-2022

GENERAL COMMENTS	I appreciate the opportunity to review the manuscript (Women's priorities towards ovarian cancer testing: A bestworst scaling study). This is a very well designed study that address an important question. The manuscript is very well written. I have few minor comments. Page 6 line 5: As Hall etal's review identified around 250 significant attributes relating to cancer testing, does that mean around 225 attributes were excluded during the Delphi process because they were not relevant to ovarian cancer testing in symptomatic? Also, is my understanding correct that the panels of the Delphi methods were the current study's authors? Page 8 line4: Were the attributes defined to participant as in Table 1? I find the study very well designed albeit the mode of questionnaire administration and recruitment method contributed to the unrepresentative sample of the study. Could the authors justify their data-collection plan (i.e. online survey and the use of Prolific for recruitment) The limitation of the study that relates to the representativeness of the sample was mentioned in the strength and limitation bullets but not in the main text.
---

VERSION 1 – AUTHOR RESPONSE

Reviewer 1

Comment: The strengths and weakness of the study are somewhat trivial and highlight concepts that done really impact this study. For example, the authors mix concepts of preferences (like predicting market uptake) with the values/priorities elicitation process inherent in a BWS. Given this is a values/priorities elicitation, issues like hypothetical bias don't apply. It would be great for the authors to focus on the substantive and practical issues associated with this application.

Response: We have sought to draw a better distinction between BWS and alternative preference methods throughout the manuscript. We have replaced references to “preferences” with “priorities” in most instances. We have also updated the strengths and limitations within the discussion (pg. 17) and key points (pg. 3). Discussion of weaknesses aim to be more specific to this study and reflect on the possible role of sample size and choice task construction on the poor distinction between mid-scale items and lower-scale items.

Comment: In the first paragraph of the introduction, which is a little terse, it would be good to know a little more about the personal and society burden of ovarian cancer.

Response: We have added to this paragraph to add further context to the burden of ovarian cancer and the importance of earlier diagnosis and testing.

Comment: Section 2.1 is a nice background, but could do with more referencing to let the reader know that this is often used method and supported by decision makers. Section 2.2 is title experimental design, but is more about identification of the factors, then experimental design – maybe this is best as two sections. Also, it is normal to call these objects, not attributes, so as not get confused with preference studies. It would be nice for the experimental design section to have some references and to more rigorously test the characteristics of the design.

Response: Thank you for these suggestions, we have implemented several changes based on this feedback.

- We have added further information on the origins of BWS and current popularity in healthcare research, complete with additional references (pg. 5, lines 10-13)
- We have separated section 2.2. into two separate sections “2.2. Identification of relevant items” and “2.3 BWS task construction”
- We have replaced the use of “attributes” to “items” or “characteristics” throughout the manuscript. We believe this differentiates from preference studies and is more suited to the readability of manuscript than “objects”.
- Additional information and references were added to the section 2.3 regarding the BIBD design (pg. 7)

Comment: “Patient and participant engagement” – this is great, but maybe need to be earlier in the methods section. It might be nice to add some references to support the standards and impetus for doing this. Also, what impact did these engagement efforts have?

Response: We have repositioned this section and added an additional sentence explaining the outcomes of the engagement (Pg 8., Lines 9-11)

Comment: Analysis – MaxDiff is one of the ways to analyze this data, but there are others. Why was this approach chosen? This is then followed by a discussion of a simple Best minus worst scale – which is also good, but different. I think a more rigorous discussion of analysis (both options and decisions) is warranted. The results don't seem to present MaxDiff scores?

Response: Thank you for this comment and clarification. Based on your feedback and further reading, we now realise this was a misunderstanding on our part due to the historic conflation of Maxdiff and BWS which we have since learnt practitioners are seeking to correct. Upon reflection it appears sequential assumption is more likely to be reflective of behavioural patterns, however, since response patterns were not directly observed we have removed any mention of Maxdiff from the manuscript and do not specify a theory of respondent behaviour within the manuscript.

Comment: Figure 2 is wonderful – but needs more explanation. What is on the Axis (I know, but the average reader may not)? What does it mean to be a cluster here? What are the hypotheses being tested? I think a more rigorous methods section is needed to explain this.

Response: Thank you. We have updated the figure to include a key which was previously missing. We have also added additional description of the figure in the results section of the manuscript (Pg 13, section 3.2.2). Again, from further reading we realise the term “clusters” may be misleading since this was based on visual inspection rather than formal cluster analysis. We have therefore removed this from the figure and replaced reference to “clustering” with the word “grouping”. Finally, the diagram was updated to reflect importance scores resulting from the updated analysis using conditional logit estimates.

Comment: Figure 3 is important, but hard to read. Also, if this was a MaxDiff or sequential best worst, then a ratio scale of importance could be made. This might really help here. Does this need to be on a bidirectional scale or unidirectional scale? Could the differences between the groups be scaled (internal consistency within the groups).

Response: We have made a number of changes based on this feedback:

- To complement the counting analysis, we introduced conditional logit analysis
- To aid interpretation, we estimated standardised importance scores, where scores sum to 100. This allowed figures 3 and 4 to be updated to a unidirectional scale, improving the readability of the figures.
- Heteroscedastic models were used to investigate the role of scale heterogeneity on observed difference between subgroups. For both subgroup analyses, the scale factor was not significant, indicating no difference in error variance between groups (Appendix 5).

Comment: Appendix 2 is nice, especially as several decision makers want heterogeneity analysis (e.g. FDA/CDRH). This said, it is not formatted great.

Response: Thank you for your comment, we have improved the layout of appendix 2 by clarifying axis labels and titles for each chart.

Comment: General discussion is good, but I would like to see the methodological issues addressed first. There are many easy fixes here and this could be a great paper with some more robust methods

and clearer presentation of the results.

Response: Thank you. As previously mentioned, the strengths and limitations section of the discussion has been updated to highlight some of the more specific issues relating to the application of BWS in this study.

Reviewer 2

Comment: Page 6 line 5: As Hall et al's review identified around 250 significant attributes relating to cancer testing, does that mean around 225 attributes were excluded during the Delphi process because they were not relevant to ovarian cancer testing in symptomatic? Also, is my understanding correct that the panels of the Delphi methods were the current study's authors?

Response: Thank you for your comment. 280 attributes in total were included in the 49 studies included within our previous review, however the number of unique attributes was much lower. The Delphi method with clear criteria for exclusion (e.g. relevance to ovarian cancer, relevance to a diagnostic context, relevance to public/patients, relevance to NHS system) was adopted to minimise bias. As you point out, ultimately this was still subject to authors judgement, with the additional input of a PPI representative. To account for this an additional question was added to the survey asking respondents if they thought any important characteristics were missing. Thirty-five respondents suggested additional attributes, although the majority of attributes had been previously excluded based on relevance to the disease (e.g. side effects, recovery time, preparation). We have not this information in the manuscript due to limit the word count.

Comment: Page 8 line4: Were the attributes defined to participant as in Table 1?

Response: Yes this is correct. We have added to the table caption to clarify this (pg 6)

Comment: I find the study very well designed albeit the mode of questionnaire administration and recruitment method contributed to the unrepresentative sample of the study. Could the authors justify their data-collection plan (i.e. online survey and the use of Prolific for recruitment)

Response: Thank you for your feedback. Prolific is widely used in both academic and industrial studies in a range of fields. We have added the following sentence to the manuscript, complete with references to justify the use of the platform within the study:

"Results from this platform have been shown to be of comparable or better quality than university research lab registers and have been used widely within hundreds of published studies across disciplines" (Pg 8, lines 21-23).

The restrictions to our sample (women, over 40, with ovaries) meant a representative sample could not be automatically implemented within Prolific. Upon examination of demographic information, it became apparent that our oversight resulted in a less representative sample. Due to the novel application of the BWS to ovarian cancer, we believe our findings still provide useful insights into the priorities of women, however, we acknowledge further research may be necessary to fully understand the generalisability of our findings.

The use of online studies have become increasingly popular in recent years. Our study was conducted during the Covid-19 pandemic meaning alternative methods such as in-person collection were not feasible.

Comment: The limitation of the study that relates to the representativeness of the sample was mentioned in the strength and limitation bullets but not in the main text.

Response: Thank you for highlighting this issue. We have updated the discussion section of the manuscript to include this limitation (pg 17. Lines

VERSION 2 – REVIEW

REVIEWER	Al-Aqeel, Sinaa recommended King Saud University , Clinical Pharmacy
REVIEW RETURNED	07-Jul-2022
GENERAL COMMENTS	The authors have addressed all comments.